# One-Layer Real-Time Optimization Using Reinforcement Learning: A Review with Guidelines

**Ruan de Rezende Faria** [1,*] , **Bruno Didier Olivier Capron** [1] , **Maurício B. de Souza, Jr.** [1,2] and **Argimiro Resende Secchi** [2]

1    Escola de Química, EPQB, Universidade Federal do Rio de Janeiro, Rio de Janeiro 21941-909, Brazil
2    Programa de Engenharia Química, PEQ/COPPE, Universidade Federal do Rio de Janeiro, Rio de Janeiro 21941-972, Brazil
*    Correspondence: rrfaria@eq.ufrj.br

**Abstract:** This paper reviews real-time optimization from a reinforcement learning point of view. The typical control and optimization system hierarchy depend on the layers of real-time optimization, supervisory control, and regulatory control. The literature about each mentioned layer is reviewed, supporting the proposal of a benchmark study of reinforcement learning using a one-layer approach. The multi-agent deep deterministic policy gradient algorithm was applied for economic optimization and control of the isothermal Van de Vusse reactor. The cooperative control agents allowed obtaining sufficiently robust control policies for the case study against the hybrid real-time optimization approach.

**Keywords:** one-layer approach; economic optimization; process control

## 1. Introduction

Real-time optimization (RTO) is based on a control system that is designed to drive the plant to reach the project decisions made in the planning and scheduling layers (i.e, optimizing the plants' economic performance as its principal objective). It is an intermediate optimization layer and performs hourly decision-making, providing the reference trajectories for the process and control variables, which hierarchically must be maintained by the supervisory control layer and, then, by the regulatory control layer, as shown in Figure 1 [1–4].

Since the 1980s, real-time optimization techniques applied to the process industry have evolved significantly. Mochizuki et al. [5] described the technological achievements that have enabled RTO to grow while simultaneously dropping in cost, which is mainly due to the development of automated and integrated optimization technologies. This is the case for refinery and chemical plants [6], especially as a consequence of the improvement of the mathematical models and of optimization packages with sufficiently robust numerical methods to optimize the economic performance of the plant and help the decision-making of engineers, who do not need to deal with the steps of data reconciliation, the updating of the model parameters, and optimization simultaneously [7].

The main challenge for RTO implementation is the integration of the layers illustrated in Figure 1. The classical two-layer approach deals with the integration of real-time optimization and supervisory control. Steady-state real-time optimization (SSRTO) demands that the plant reach a steady-state for the optimization with a rigorous model to be performed. Until this condition is satisfied, nothing can be done [8]. Model predictive control (MPC) implements control actions in this interval (minute by minute), using a simpler model that captures the process dynamics and drives the design variables to their optimal values. However, this integration can be complex when implementing the set-point resulting from SSRTO because the plant must still be in the original steady-state. In addition, the mismatch model issue must be treated so that the high-level SSRTO sends

admissible set-points for the lower-level MPC [8–11]. Dynamic real-time optimization (DRTO) is an alternative derived from SSRTO, which requires a rigorous dynamic model of the process to eliminate steady-state detection requirements. However, implementing it for large-scale systems is challenging (i.e., regarding its modeling and optimization), even with the computational power currently available [12,13]. A more recent approach combines SSRTO and DRTO, called hybrid real-time optimization (HRTO), which has the modeling effort reduced because the dynamic terms in the model need only to be introduced in the model adaptation step, thus reducing the steady-state waiting time (e.g., [13–16]).

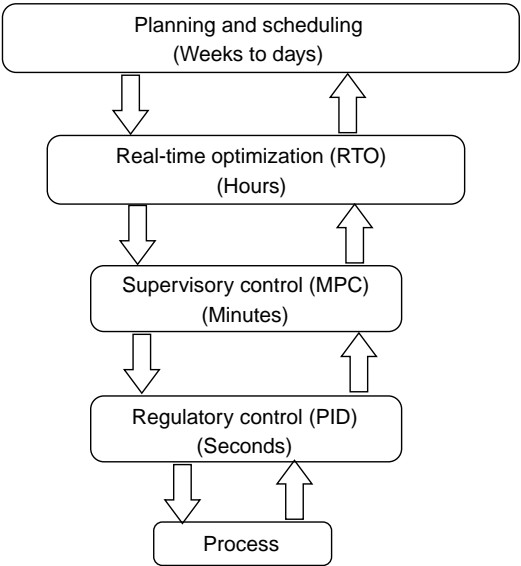

**Figure 1.** The typical control and optimization system hierarchy in industry.

Alternatively, the one-layer approach has been attempted to circumvent these problems. The objective is to augment the MPC with global plant information (e.g., plant economic objective) to remove the SSRTO layer. Zanin et al. [17,18] applied this methodology to fluid catalytic cracking units, using MPC with an economic factor directly included in the control objective. These studies were precursors for developing a new line of research called economic model predictive control (EMPC) [19,20]. Its main advantage is derived from a controller with better performance in terms of disturbance rejection and economic profits, provided that the control tuning has been correctly performed and the security constraints included. However, the computational burden for the one-layer approach may be huge, especially when it is solved via a global optimization algorithm due to the nonlinearity and non-convexity of the problem [21].

A machine learning approach called reinforcement learning (RL) is showing promising results in several areas of knowledge mainly linked to artificial intelligence (e.g., natural language processing [22], autonomous driving [23], robotics [24,25]). Due to the remarkable results obtained in these areas, which are a consequence of the consolidation of deep neural networks and new reinforcement learning algorithms, this methodology effectively began to be studied by the process control community (e.g., [26–30]). It is characterized by an agent (i.e., control policy) capable of self-learning in the process guided by numerical rewards, following a Markov decision process. Specifically, the agent learns from interactions without relying on a process model (i.e., model-free), and data-driven and simulation information can be used. Thus, reinforcement learning is considered as a promising alternative to replace or complement standard model-based (MPC) approaches for batch process control (e.g., for more details, see [31,32]). At this point, the present article reviews the two-layer and one-layer real-time optimization approaches from the point of view of reinforcement learning. The contribution to the state-of-the-art is twofold: (1) There is a lack of articles covering the subject. To the authors' knowledge, only Powell et al. [33] implemented RL for the real-time optimization of a theoretical chemical reactor. (2) The research method

involves the study of each online optimization layer of the plant-wide structure, explicitly dealing with its conceptualization for reinforcement learning, as well as the challenges for integrating the layers and the problems of the implementation and maintenance of control agents under this point of view.

This review begins in Section 2 with a brief introduction to the reinforcement learning methodology. Section 3 details real-time optimization, supervisory control, and regulatory control for RL. Section 4 proposes a benchmark study of RL using a one-layer approach, evaluating the computational burden and control performance when the disturbance varies with the same plant dynamics, which is a challenge independent of the employed RTO approach. For that purpose, the multi-agent deep deterministic policy gradient algorithm is applied for economic optimization and control of the isothermal Van de Vusse reactor.

## 2. Reinforcement Learning

The reinforcement learning concept is briefly given in Figure 1. In terms of defining the elements of RL, this self-learning process comprises the interaction of an agent through actions ($a_t$) with the environment, which reaches a new state ($s_{t+1}$) guided by a reward ($r_t$). The first classic example goes back to the theory of animal psychology, exemplified by an animal (i.e., agent) learning how to perform a certain task in a controlled environment, with the reinforcement signal used to guide its learning (e.g., rewards in the form of food) [34]. With the advent of programmable computers, reinforcement learning theory intersected with artificial intelligence. Minsky [35] discussed RL models. Bellman [36,37] founded the theory of optimal control, dynamic programming, and the Markov decision process. Besides these advances, the problems related in Marvin and Seymour [38] about the perceptron network affected the state-of-the-art development in artificial intelligence until the mid-1980s.

With the return of interest in artificial intelligence (e.g., with the work of Rumelhart et al. [39] as a landmark), classic RL problems began to be studied again. For example, the pole balancing problem (a benchmark optimal control problem) provided RL-based alternatives to black-box models obtained from neural networks and tabular algorithms. Figure 2 outlines this learning problem, where an agent must keep the pole balanced (when pushing the cart to the right or left) for a defined length of time of the simulation (*T*), being formulated as a stochastic sequential decision-making problem (more details are given in Section 2.1), in which the optimal policy must maximize the reward sum, known as the return $R(\tau)$, along the trajectory $\tau = (a_0, s_0, a_1, s_1, \cdots, a_T, s_T)$, as shown in Equation (1). In this equation, $\gamma$ corresponds to the discount factor of the return. When $\gamma = 0$, the immediate reward ($r_t$) is prioritized, whereas if $\gamma = 1$, the entire trajectory is considered [40–42].

$$R(\tau) = \sum_{t=1}^{T} \gamma^{t-1} r(s_t, a_t, s_{t+1}) \tag{1}$$

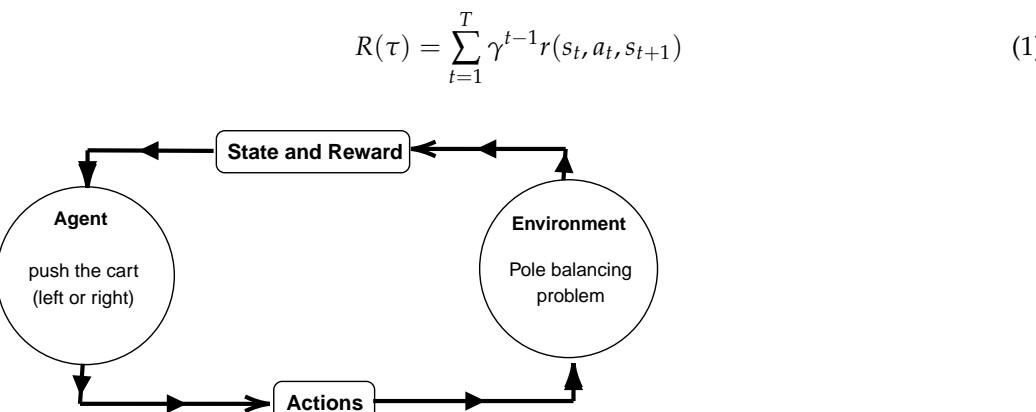

**Figure 2.** Simplified outline of the pole balancing problem in the RL framework.

### 2.1. Markov Decision Process

The Markov decision process (MDP) is defined by the tuple $(S_t, A_t, P_t, R_t)$. $S_t$, $A_t$, and $R_t$ correspond to the set of non-terminal states, actions, and rewards. $P_t$ is the probability transition, with $P_t : S_t \times A_t \times S_t^+ \to [0, 1]$ and $S_t^+$ the set of terminal states. As mentioned in Section 2, the MDP is a discrete-time stochastic control process, in which the current state of the system ($s_t \in S_t$) must contain all the information needed by the agent to decide which action ($a_t \in A_t$) to take (independent of all previous states and actions), resulting in the subsequent transition to the new state $s_{t+1} = P(s_{t+1}|s_t, a_t)$, in order to satisfy the Markov property. Moreover, a policy ($\pi$) must map states to actions and is optimal when it maximizes the return regardless of the initial state chosen ($s_1$) and the obtained trajectory ($\tau$) (Equation (2)) [42,43].

$$\pi^* = \underset{\pi}{argmax} E_{p^\pi(\tau)}[R(\tau)] \tag{2}$$

In this equation, $E_{p^\pi(\tau)}$ denotes the expectation about the trajectory $\tau$ extracted from $p^\pi(\tau)$, and $p^\pi(\tau)$ denotes the probability density of observing the trajectory $\tau$ under policy $\pi$ (Equation (3)).

$$p^\pi(\tau) = P(s_1) \prod_{t=1}^{T} P(s_{t+1}|s_t, a_t) \pi(a_t|s_t) \tag{3}$$

### 2.2. Algorithms

The main reinforcement learning algorithms can be divided into two main categories [40]. Value-based algorithms obtain the optimal policy by approximating the return value for all possible trajectories. The first proposed alternative considers only the state value function (Equation (4)). The other option considers the value function of the state–action pair (Equation (5)) [41].

$$V^\pi(s) = E_{p^\pi(\tau)}[R(\tau)|s_1 = s] \tag{4}$$

$$Q^\pi(s, a) = E_{p^\pi(\tau)}[R(\tau)|s_1 = s, a_1 = a] \tag{5}$$

An algorithm consolidated in the literature is known as Q-learning and was first described by Watkins [41]. This outstanding work was a watershed for reinforcement learning theory. The main outcome of this study was to replace the tabular methods with parametric approximators to improve generalization and deal with the dimensionality problem. The optimization problem now conforms to Equation (6), and the agent explores the environment until the Bellman optimality condition is reached (i.e., using bootstrapping) (Equation (7)).

$$\theta^* = \underset{\theta}{argmin}[Q^\pi(s, a, \theta) - Q^\pi(s, a)] \tag{6}$$

$$Q^\pi(s, a) = r(s, a) + \gamma Q^*(s', a', \theta) \tag{7}$$

The second category includes the policy-based algorithms. Williams [44] developed the first algorithm based on these principles, which is known as REINFORCE and has shown promising results in several fields of research (e.g., [29,30]). Specifically, it uses an auto-parameterized policy ($\pi(a_t|s_t, \theta)$) and should maximize the expected return $J(\theta)$, as shown in Equations (8)–(10).

$$\theta^* = \underset{\theta}{argmax} J(\theta) \tag{8}$$

$$J(\theta) = E_{p^\pi(\tau|\theta)}[R(\tau)] = \int p(\tau|\theta) R(\tau) dh \tag{9}$$

$$p(\tau|\theta) = p(s_1) \prod_{t=1}^{T} p(s_{t+1}|s_t, a_t) \pi(a_t|s_t, \theta) \tag{10}$$

### 2.3. Deep Reinforcement Learning

Section 2.2 briefly provided the main RL algorithms described in the literature. There is a consensus that deep neural networks are the best option as parametric approximators due to their ability to deal with big data in terms of generalization, computational processing power, and the invariance dilemma [45]. Another advance concerns the development of stable algorithms to deal with problems demanding high-dimensional state space evaluation and alternatives to improve the offline training of these agents (e.g., [46–48]).

The most-employed deep reinforcement learning algorithm in the literature is the actor–critic networks. This is an algorithm that combines ideas from value-based and policy-based algorithms. First, it was described in LeCun et al. [49]. Then, Sutton et al. [50] effectively formulated it for RL, with the guarantee of convergence given by the deterministic policy gradient theorem. Figure 3 illustrates the update of the actor and critic networks. Specifically, the step forward comprises the action selection ($a = \pi(a_t|s_t, \theta_a)$) and the critic network computation (i.e., value function $Q(s_t, a_t, \theta_c)$). In the backward step, the parameters of both networks are updated with the backpropagation and stochastic gradient descent algorithms. At this point, it is worth mentioning that it is necessary to calculate $\nabla_a \log(\pi(a_t|s_t, \theta_a))$ since the actor network categorizes actions instead of their value, as suggested by Williams [44] and shown in Equation (10). Furthermore, the temporal difference (TD) method is recommended to learn offline directly from experience (i.e., TD-error ($\delta_t$)). As the method does not require a model of the environment, it is straightforward to implement and allows bootstrapping, which is an advantage compared to Monte Carlo (MC) methods and dynamic programming. However, its training must be rigorously performed to avoid obtaining unfeasible policies, since the computed Q-value does not consider the return from the entire trajectory as in MC methods [51].

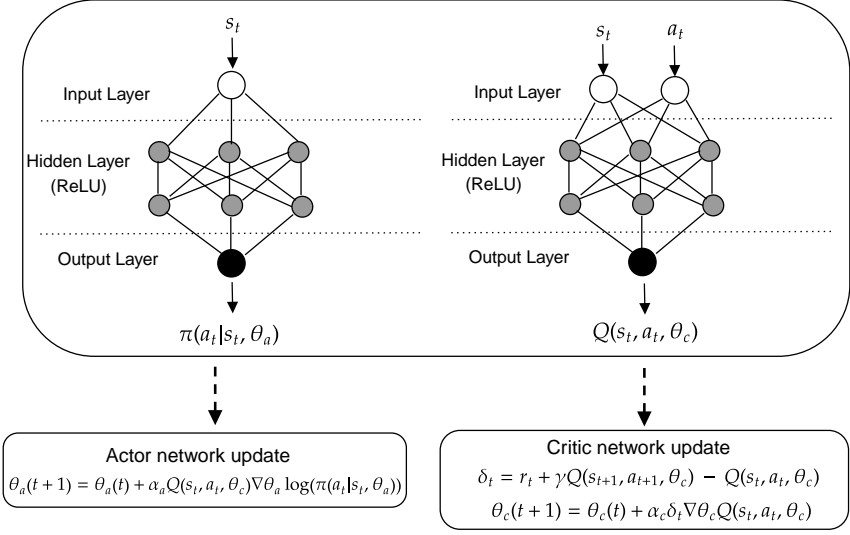

**Figure 3.** Diagram describing the update of the actor and critic parameterized by a deep neural network.

This model-free algorithm depends on offline training, where the learning across different episodes can be: (1) off-policy: the learned policy is updated from data obtained from the implementation of other policies. This learning procedure is shown in Figure 4, where $\pi_{t+1}$ is updated from samples of policy roll-out data up to $\pi_t$ (i.e., $\pi_1, \cdots, \pi_t$), which are contained in a large buffer. For example, the deep deterministic policy gradient (DDPG) algorithm [47] estimates the Q-values through a greedy policy instead of the behavioral policy, which leads to the TD-error ($\delta_t$) shown in Equation (11). The second (2) is on-policy: the learned policy $\pi_{t+1}$ is updated from transitions (or data) exclusively taken from the previous policy $\pi_t$; thus, it does not depend on a buffer [42]. For example, the proximal policy optimization (PPO) algorithm [52] estimates the Q-values assuming

the current behavioral policy continues to be followed, resulting in the TD-error ($\delta_t$) shown in Equation (12).

$$\delta_t = r_t + \gamma\, max\, Q(s_{t+1}, a_{t+1}, \theta_c) - Q(s_t, a_t, \theta_c) \tag{11}$$

$$\delta_t = r_t + \gamma Q(s_{t+1}, a_{t+1}, \theta_c) - Q(s_t, a_t, \theta_c) \tag{12}$$

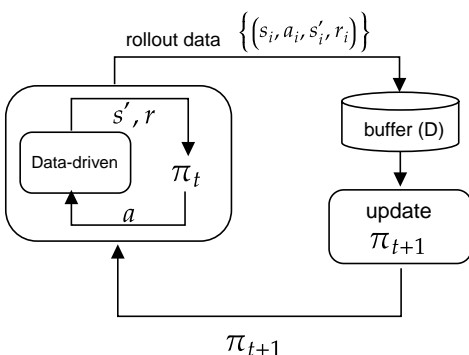

**Figure 4.** Off-policy reinforcement learning.

## 3. Applications

This section details the specific applications of reinforcement learning for real-time optimization, supervisory control, and regulatory control. The main details of these methodologies are also presented. Table 1 summarizes several references for RL in this context, in which the listed algorithms are derived from those shown in Section 2.

**Table 1.** Several references for reinforcement learning.

| Main Topic | Algorithm | References |
|---|---|---|
| SSRTO | Deep actor–critic | [33] |
| Supervisory control | REINFORCE | [29,30] |
| | Deep Q-learning | [53] |
| | PPO | [28] |
| | DDPG | [26] |
| | DDPG | [54] |
| | A2C | [55] |
| Regulatory control | Deep Q-learning | [56] |
| | PPO | [57] |
| | DDPG | [58] |
| | DDPG | [59] |
| | A3C | [27] |

The deep deterministic policy gradient algorithm is actor–critic, off-policy, and deterministic ($\mu(s_t, \theta_t)$). This algorithm is an improved version of the deterministic policy gradient algorithm for continuous control. The critic is updated using the Bellman equation (Equation (13)). The actor is updated by applying the chain rule to the expected return (weighed by the critic) with respect to the actor parameter (Equation (14)) [46,47].

$$\delta_t = E\left[r_t + \gamma\, max\, Q(s_{t+1}, a_{t+1}, \theta_t) - Q(s_{t+1}, a_{t+1}, \theta_{t+1})\right]$$
$$\theta_{t+1} = \theta_t + \alpha_c \delta_t \nabla_{\theta_t} Q(s_t, a_t, \theta_t) \tag{13}$$

$$\theta_{t+1} = \theta_t + E\left[\nabla_{\theta_t} \mu(s_t, \theta_t) \nabla_{a_t} Q(s_t, a_t, \theta_t)\right]\big|_{a_t = \mu(s_t, \theta_t)} \tag{14}$$

The A2C and A3C algorithms [60] are variations of the actor–critic algorithm with agents learning asynchronously, with two or three agents in parallel. The only exception is the proximal policy optimization algorithm, which is an algorithm that learns while interacting with the environment over different episodes (i.e., on-policy). Methodologically, this property comes from another similar algorithm considered more complex (trust region

policy optimization (TRPO)), addressing the divergence effect Kullback–Leibler (KL) and surrogate objective functions. Based on this, what is summarized in Table 1 is consistent with the works reported in [61–63], with a preference for DDPG and PPO algorithms and their variants as learning algorithms for process control.

### 3.1. SSRTO

In this review, SSRTO is the only RTO application addressed for three reasons: (1) it remains the most-applied approach in the process industry; (2) it has an extensive literature; (3) to the best of the authors' knowledge, only Powell et al. [33] used RL in this context. Figure 5 illustrates the steps of SSRTO with model parameter adaptation (MPA). Specifically, MPA updates the rigorous model of the SSRTO layer with steady-state plant information, which must be detected and treated with process data reconciliation before being fed to the parameter estimator. The process disturbances and model uncertainty strongly influence this step. When varying with the same plant dynamics, the former makes it difficult to detect the steady-state. The latter directly affects data reconciliation and parameter estimation. Nevertheless, it is a traditional industry methodology for well-segmented optimization and control steps, facilitating its application and maintenance in the real process. In addition, there are still alternatives to improve parameter estimation by including model and plant derivatives (for more details, see [64–66]).

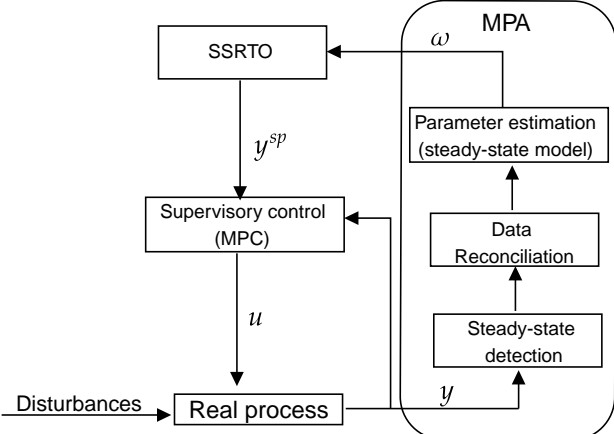

**Figure 5.** The classical two-layer approach in the industry. The steps of SSRTO with model parameter adaptation and model predictive control.

For brevity, the optimization details about the MPA approach and SSRTO layer are shown in Equations (15) and (16), respectively, provided that the steady-state detection and data reconciliation steps have been correctly performed. In Equation (15), the parameters of the rigorous model of the process (i.e., $y = f(u, \omega_k)$) are updated ($\omega_k^*$) based on steady-state reconciled measurements from the plant ($y_{meas}$), with the values of the manipulated variable ($u$) received from the supervisory control. In Equation (16), the updated model is used to optimize the plant encompassing the process and economic constraints ($g(y, u)$). The optimal solution $Y_{k+1}^* = (y_{k+1}^*, u_{k+1}^*)$ is sent as the set-point to the supervisory control layer, which must drive the plant to these desired values, while rejecting process disturbances [11,13].

$$\omega_k^* = \underset{\omega}{argmin}||y_{meas} - f(u, \omega_k)|| \tag{15}$$

$$Y_{k+1}^* = \underset{u,y}{argmin} J(y, u)$$
$$s.t. \; y = f(u, \omega_k^*) \tag{16}$$
$$g(y, u) \leq 0$$

Considering RL in the MPA framework, the reformulated problem now comprises offline training, policy transfer, and online policy deployment. Moreover, the elements of RL encompass the current state of the plant $s_k = f(u, \omega_k)$, action $a_k = \omega_{k+1}$, and reward $r_k = ||y_{meas} - f(u, \omega_k)||$. First, extensive offline training based on the randomization of process variables and parameters is required to sample $a_k$, $s_k$, $s_{k+1}$, and $r_k$. To deal with the complexity of offline training, adopting a feasible range for the parameters (i.e., defining the lower bound $lb < a_k$ and upper bound $a_k < ub$) and using a replacement model of the rigorous model of the process based on deep neural networks are recommended, which limits the agents' exploration and exploitation of it. As a result, this allows an adaptive, stochastic and low-computational-cost policy, depending only on the computation of the action by the actor network ($\omega_{k+1} = \mu(s_k, \theta_a)$). At the end, policy transfer and online policy deployment remain to be addressed by RL. For that purpose, safe RL can guarantee its industrial application in the future (see, for instance, [28]).

SSRTO using RL was addressed by Powell et al. [33], who proposed a methodology to deal with the economic objective and process constraints for a chemical reactor control problem (a continuously stirred tank reactor (CSTR)). Specifically, they simulated data from a rigorous process model. They included process constraints in the reward and the price of reagents and loads ($S$) as the input to the actor and critic networks. The RL elements encompass the current plant state $s_k = f_k(a_k, \omega_k)$, action $a_k$, and reward $r_k = J(s_k, a_k, S)$. For the offline training phase, the vanilla actor–critic algorithm was employed. The training was divided into two stages: (1) the critic network was trained based on the randomization of the process conditions and using cross-validation to obtain an unbiased and generalizing critic network (i.e., $Q(s_k, a_k, S, \theta_c)$); (2) the actor network $\mu(s_k, S, \theta_a)$ was trained around the optimal operating conditions of the plant and the feasible range of the parameters, taking $S$ as known disturbance and adding constraints for the actions, as shown in Equation (17), with $\lambda$ adjusting the importance of the magnitude of the action.

$$a_k^* = \underset{a_k}{argmax}\ Q(s_k, a_k, S, \theta_c) + \lambda(a_k)$$

$$s.t.\ a_k = \mu(s_k, S, \theta_a)$$

$$\lambda(a_k) = \begin{cases} -1000, & \text{if} \quad a_k < lb_a \\ -1000, & \text{if} \quad a_k > ub_a \\ 0, & \text{Otherwise} \end{cases} \tag{17}$$

After extensive offline training, Powell et al. [33] applied the policy to the real process (i.e., adding white noise and altering S in the simulated process). As a result, the control law was adequate to what would be required by an operator for automatic control or supervision, demanding the evaluation of a function instead of solving the RTO problem. Despite this, the obtained control law generated a smaller economic gain than SSRTO solved by nonlinear programming.

### 3.2. Supervisory Control

This section focuses on RL's applications for supervisory control, rather than describing the implications for the development of control theory itself (what can be seen in [62,63]). In Table 1, the listed references comprise RL methodologies applied to the batch process control due to the scarce literature on continuous process control, which mainly combines reinforcement learning and model predictive control to include process constraints and, thus, ensure a level of control stability. Therefore, they guide the discussion throughout this section.

For example, Ma et al. [26] formulated the control problem of a semi-batch polymerization reactor. For MDP sampling (regarding the current process state, action, and reward), extensive offline training was conducted so that the optimal set-points were randomly selected and modified during off-policy learning and the zero-mean Ornstein–Uhlenbeck process was applied to the actions to generate temporally correlated exploration samples

(as in the DDPG algorithm). Additionally, the state space was increased by including the difference between the sampled state and the desired set-point (i.e., $y_t - y_t^*$). These changes directly influence the obtained reward, as shown in Equation (18), which depends on the time $t$ and parameters $\alpha$, $\beta$, and $c$ to adjust the importance of reaching the set-point ($y_t^*$). To a certain extent, the redefinition of this control problem is close to the approach used for MPC, in which the offline training step would be the model identification. Thereafter, the controller is implemented in a closed loop considering the process constraints and the previously identified model (e.g., [67,68]).

$$r_t = \begin{cases} \alpha t + c, & \text{if} \quad ||y_t - y_t^*|| \leq 0.05 y_t^* \\ \beta ||y_t - y_t^*|| - \alpha t, & \text{Otherwise} \end{cases} \tag{18}$$

The other listed references and control applications (Table 1) are briefly commented on as they follow an RL methodology similar to the one described above, changing only the type of algorithm used, the addressed control problem, and the definition of the RL elements. Namely, all RL applications depend on extensive offline training based on simulation or real plant data, where the set-point is sampled randomly and has to be reached during the training. Another point for discussion is the lack of validation for industrial processes, as all case studies were restricted to control experiments based on simulation. However, the obtained policies showed superior performance to the model-based ones for all cases (e.g., MPC), which is promising and encourages developing state-of-the-art technologies (RTO using RL). Undoubtedly, safe RL will be essential to ensure such integration in the short term.

Based on this, a parallel is made for continuous processes. Extensive offline training based on the randomization of process conditions is also required to allow MDP sampling. However, a less rigorous model of the process is employed (when compared to that used for SSRTO), and generating simulation data for offline training is costly and complex because the optimal values of the process and control variables need to be maintained minute by minute instead of hourly as in the SSRTO layer. For example, Oh et al. [69] discussed the integration of reinforcement learning and MPC to ensure online policy implementation and fix these issues, suggesting a new approach to Equation (1) (Equation (19)), where the terminal cost is approximated by Q-learning and the cost stage by MPC (e.g., [68,70,71]). This blended receding horizon control approach allows for including constraints on controls and states and incorporating disturbances directly into the optimization problem. It implied in a controller data sampling efficiency much higher than that of model-free RL algorithms, but the challenge is executing online policies continuing to learn about the process and balancing safety and performance.

$$a^* = \underset{a(0),\cdots,a(p-1)}{argmin} \sum_{k=0}^{p-1} \left[ L(s(k), a(k)) + Q(s(p), a(p), \theta_c) \right]$$
$$s.t.\ s_{k+1} = f(s_k, a_k, \omega_k), \quad s(0) = s_0$$
$$a(k) \in A, i = 0, \cdots, p-1$$
$$s(k) \in S, i = 1, \cdots, p-1 \tag{19}$$
$$s(p) \in Z, i = 0, \cdots, p-1$$

### 3.3. Regulatory Control

The regulatory control layer is directly in contact with the process. It makes second-scale control decisions to indirectly optimize the plants' economic performance, depending on the optimal values (for the design variables) from the supervisory control layer. For regulatory control, RL can be understood as an adaptive controller, which is similar to the proportional–integral–derivative (PID) controller (for more details on PID control methods, see Kumar et al. [72]), automating the control decisions while depending on the evaluation of a function (i.e., deep neural network) [62]. Due to these remarkable features and a broad

portfolio of bench-scale and industrial-scale control experiments using PID controllers, RL-adapted alternatives also began to be proposed for bench-scale control experiments. The control policy is less conservative and can automate controller tuning to adapt to supervisory control set-point changes.

For example, Lawrence et al. [59] studied an experimental application for PID tuning. They innovated by embedding PID in the RL framework, updating the actor and critic networks (i.e., employing the DDPG algorithm) by directly using PID tuning parameters as parameters of the actor itself (i.e., proportional gain ($k_p$), integral gain ($k_i$), and derivative gain ($k_d$)) such that $a_t = \mu(s_t, \theta_a = (k_p, k_i, k_d)) + a_{t-1}$. They also increased the state space by including time-delayed information on the control action and the difference between the sampled state and the desired set-point (for more details on including historical information for Markov state prediction, see Faria et al. [32]). The authors used this difference as a reward signal and penalized actions with high-magnitude variations (i.e., $r_t = ||y_t - y_t^*|| - 0.1||\Delta u||$). Finally, considering a two-tank system-level control experiment, they compared the results to various tuning parameters (PID controller) based on the internal model control approach and evaluated the nominal performance and stability, among other factors that influenced the response of both controllers. The results showed that the RL-based controller proved efficient for such performance criteria and could follow the set-point and reject perturbations.

The above approach is an extension of the ones seen in the works of Dogru et al. [27] and Ramanathan et al. [56]. Both adapted the control experiment in the RL framework using a more conventional approach than the one proposed by Lawrence et al. [59], similar to the one used for supervisory control. Specifically, instead of updating the PID controller parameters included in the actor network, the actor and critic learn following the approach illustrated in Figure 3. In addition, both increased the state space by including information between the sampled state and the desired set-point (delayed in time) and the immediate reward penalizing large set-point deviations and large magnitude variations of actions. Spielberg et al. [58] focused on the design of the RL agent for regulatory control and rigorously detailed the steps for its implementation considering a control experiment in a simulated environment (using the DDPG algorithm). The resulting control agent followed the set-point for single-input, single-output (SISO) and multiple-input, multiple-output (MIMO) cases. Moreover, a new control experiment evaluated the robustness of the DDPG controller to adapt to abrupt process changes and how this affects its learning performance and convergence. The result was that the controller continues to learn online without the need to restart the offline training.

## 4. Benchmark Study of Reinforcement Learning

### 4.1. Offline Control Experiment

To consolidate what was described in this review, a control experiment details the formulation of a control agent based on RL. The proposed control experiment follows an example from the work of Ławryńczuk et al. [73], which evaluated the classical two-layer approach (i.e., RTO plus MPC) for the economic optimization and control of the well-known Van de Vusse reactor.

### 4.1.1. Dynamic Model of the CSTR

A schematic diagram of the Van de Vusse reactor is shown in Figure 6. The feed inflow contains only cyclopentadiene (component $A$) with concentration $C_{af}$. The volume ($V$) is maintained constant throughout the reaction. The outflow includes the remainder of cyclopentadiene, the product cyclopentenol, and two unwanted by-products, cyclopentane-diol (component $C$) and dicyclopentadiene (component $D$), with concentrations of $C_a$, $C_b$,

$C_c$, and $C_d$ and constant reaction rates ($k_1, k_2, k_3$), characterizing the Van de Vusse reaction (Equation (20)) [73].

$$A \xrightarrow{k_1} B \xrightarrow{k_2} C$$
$$2A \xrightarrow{k_3} D \tag{20}$$

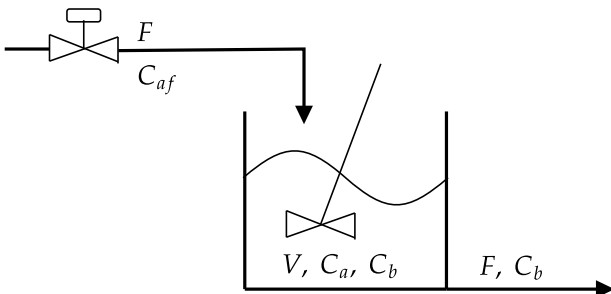

**Figure 6.** Van de Vusse reactor.

This process is modeled as an ordinary differential equation (ODE) system with two states ($C_a$ and $C_b$) and one manipulated variable ($F$), as shown in Equations (21) and (22).

$$\frac{dC_a(t)}{dt} = -k_1 C_a(t) - k_3 C_a^2(t) + \frac{F(t)}{V}(C_{af}(t) - C_a(t)) \tag{21}$$

$$\frac{dC_b(t)}{dt} = k_1 C_a(t) - k_2 C_b(t) - \frac{F(t)}{V} C_b(t) \tag{22}$$

The real-time optimization problem deals with the maximization of the concentration of the component $B$ ($J = -C_b^{ss}$) by manipulating $F \in (0, 150)$. It is also assumed that the product should fulfill a purity criterion ($C_b > 1.15$) regardless of the values of disturbance $C_{af} \in (9, 11)$. Within the classic RTO perspective (Figure 5), $w_k^*$ corresponds to $C_{af}$, the CSTR model ($y = f(u, w_k^*)$) is the same for SSRTO and MPC, and the optimal solution ($C_b(k+1)^*, F(k+1)^*$) is sent as the set-point to the supervisory control layer (MPC). In the next section, this problem is formulated in the RL framework, according to the one-layer approach. The option is to use a variant of the DDPG algorithm as a learning agent (i.e., multi-agent deep deterministic policy gradient (MADDPG) [74]), which uses cooperative or concurrent control agents to improve the data distribution (i.e., buffer sampling) and to stabilize training, which is a contribution to the literature, following the recommendations of Powell et al. [33] on obtaining an optimal actor and critic given the complex offline training.

4.1.2. RL Framework

The roll-out data sampling was based on the simulation of the Van de Vusse reactor over $k = (1, \cdots, N)$ episodes of size $i = (1, \cdots, T)$. The RL elements in this framework are illustrated in Figure 7. The current state and the new state are $x = (C_a(t), C_b(t))$ and $x' = (C_a(t+1), C_b(t+1))$. The reward is the essential RL element to guide offline learning. Its definition encompasses the economic objective ($J(t) = -C_b^{ss}$) according to Equation (23), which is maximized only when the required minimum production of $C_b$ is reached (i.e., $C_b \geq 1.15$), plus the contribution of $\eta(T - i)\Delta t$, with $\eta$ adjusting the importance of reaching such a condition as quickly as possible. When the minimum $C_b$ production is not reached, $\beta$ adjusts the importance of maximizing $C_b(t+1)$ and $C_b(t)$ and $\phi i \Delta t$ penalizes high variations of $\Delta F = \mu(x, \theta_a)$ (i.e., in the range $F \in (0, 150)$). Additionally, the reactor initial conditions are randomly selected according to continuous uniform distribution (i.e., $C_a(0) = U(a, b), C_b(0) = U(a, b)$, where $a$ and $b$ are the minimum and maximum values), with process disturbance also sampled randomly ($C_{af}(0) = U(a, b)$). They are maintained

constant for each episode, and the procedure repeats until the buffer is completely filled ($N = D$).

$$r_t = \begin{cases} \eta(T-i)\Delta t + F(t), & \text{if} \quad C_b(t) \geq 1.15 \\ \beta C_b(t+1)\, C_b(t) - \varphi ||\Delta F(t)||\,(i)\, dt, & \text{Otherwise} \end{cases} \tag{23}$$

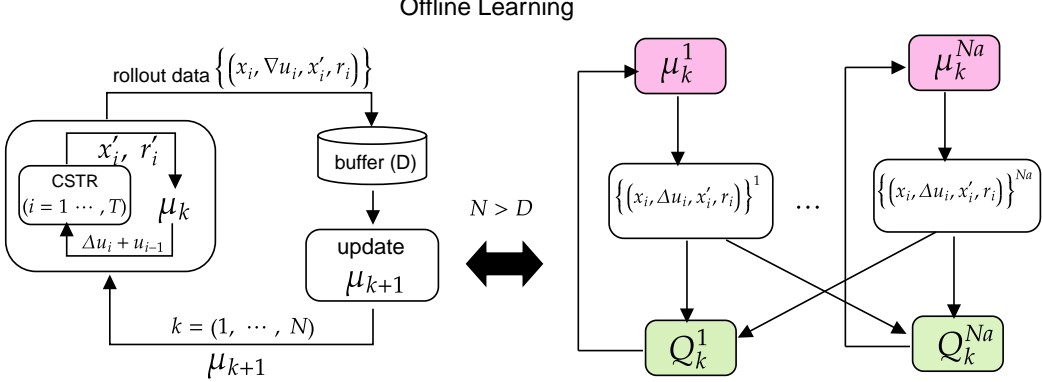

**Figure 7.** Offline learning with multi-agent deep deterministic policy gradient, with a buffer of size $D$, to optimize and control the Van de Vusse reactor.

　　　The steps for its implementation are similar to the DDPG algorithm and are shown in Algorithm 1. However, the dimension of the buffer is increased since the critic network update ($Q_1$) will also depend on collected experiences, actions $U = \{u_1, \cdots, u_{Na}\}$, states $X = \{x_1, \cdots, x_{Na}\}$, state transitions $X' = \{x'_1, \cdots, x'_{Na}\}$, and rewards $R = \{r_1, \cdots, r_{Na}\}$, from the other agents. Regarding the case study, the first step is selecting the number of agents ($\mu_t^i$), which directly influences the amount of acquired experience and the computational burden. At each execution of the algorithm (episodes $= 1, N$), the initial condition ($X_0$) is sampled at random (uniformly) for each agent and kept constant for $t = 1, T$. The aim is to expose each agent to various experiences, continuing to learn to maximize rewards ($R_t$) regardless of the initial state ($C_a(0)$ and $C_b(0)$) and process disturbance ($C_{af}(0)$). This procedure follows the behavioral policy and the Ornstein–Uhlenbeck process to generate temporally correlated exploration samples ($u_t^i = \Delta F_t^i$), and thus obtain the rewards (Equation (23)) and reach the new state (Equations (21) and (22)), which compose a buffer that stores and replays uniformly the roll-out data ($\{X_t, U_t, R_t, X_{t+1}\}$). With a sufficient number of samples (i.e., if $N > D$), the update step for each agent effectively begins. The tuple ($\{X_j^i, U_j^i, R_j^i, X_{j+1}^i\}$) of size K is sampled at random (uniformly) to compute the TD-error ($\delta_j^i$) increased with actions implemented by each agent, which leads to updating the critic and actor networks. Finally, updating the actor and critic networks with delayed (or filtered) copies of the original DNN is an alternative, as in DDPG algorithms, to stabilize training.

---

**Algorithm 1:** MADDPG algorithm.

---

    **Result:** For each agent $i$, optimal policy $\mu^i(x_t^i, \theta_a^i)$

1  **for** *episodes* $= 1, N$ **do**

2     $X_0 = (C_a(0), C_b(0), C_a f(0))$ e $U_0 = F(0)$;

3     $\xi$ is a zero-mean Ornstein–Uhlenbeck process;

4     **for** $t = 1, T$ **do**

5         For each agent $i$, select actions $u_t^i = \Delta F_t^i = \mu^i(x_t^i, \theta_a^i) + F_{t-1}^i + \xi_t$;

6         Obtain reward $R_t$ and new state $X_{t+1}$;

7         Replay buffer $\{X_t, U_t, R_t, X_{t+1}\}$;

8         **if** $N > D$ **then**

9             **for** $i = 1, N_a$ **do**

10                **for** $j = 1, K$ **do**

11                   roll-out data sampling $\{X_j^i, U_j^i, R_j^i, X_{j+1}^i\}$;

12                   Compute the temporal difference error ($\delta_t$):

$$y_j^i = r_j^i + \gamma \, Q_{\theta_c'}^i(X_{j+1}^i, \mu^i(x_{j+1}^1, \theta_a'^i), \cdots, \mu^i(x_{j+1}^{N_a}, \theta_a'^i));$$

13                   $\delta_j^i = y_j^i - Q_{\theta_c}^i(X_j^i, u_j^1, \cdots, u_j^{N_a});$

14                **end**

15              Critic network update:

16              $L_c^i = (\frac{1}{K} \sum \delta^2)^i;$

17              Actor network update:

18              $\nabla_{\theta_c}\mu|_{x_i} \approx \frac{1}{K} \sum_i \nabla_a Q_{\theta_c}^i(X_t^i, \mu^i(x_t^1, \theta_a^i), \cdots, \mu^i(x_t^{N_a}, \theta_a^i)) \, \nabla_a \mu^i(x_t^i, \theta_a^i);$

19             **end**

20             For each agent i:

21             Target critic network update: $\theta_c^i(t+1) = \kappa\theta_c^i(t) + (1-\kappa)\theta_c^i(t+1)$ ;

22             Target actor network update: $\theta_a^i(t+1) = \kappa\theta_a^i(t) + (1-\kappa)\theta_a^i(t+1)$

23         **end**

24     **end**

25 **end**

---

4.1.3. Validation of the Control Experiment

The parameters used in the offline control experiment are summarized in Table 2. The selected hyperparameters' values (i.e., $K$, $\kappa$, and $\gamma$) followed the recommendations of Lillicrap et al. [47] and Lowe et al. [74]. The number of episodes ($N$), the buffer size ($D$), the number of agents ($Na$), the size of the episode time horizon ($T$), and the parameters $\eta$, $\beta$, and $\varphi$ were selected by trial and error. The selected actor network must be more conservative than the critic network, according to the recommendations from Faria et al. [32]. Furthermore, the actor and critic learning rates decay by $\vartheta$ every 1000 episodes (i.e., when $N \geq D$) to stabilize stochastic gradient descent optimization. The definition of the process conditions, except for the control action limit ($\Delta F$) between -20 and 20, followed the example of Ławryńczuk et al. [73]. Namely, $C_a(0) = U(9, 11)$, $C_b(0) = U(0, 1.5)$, $C_{af}(0) = U(9, 11)$, $F(0) = U(0, 150)$, $k_1 = 50$, $k_2 = 100$, $k_3 = 10$, and the control experiment lasting two hours, with the control action taken at each time interval of duration of 0.025 h ($\Delta t = 0.025$), totaling 80 time steps (i.e., $T = \frac{2}{0.025} = 80$). Finally, the control experiment was performed on an ACER Aspire A315-23G computer with 12 GB RAM. The system of Equations (21) and (22) was integrated using CVODES [75] (suite of differential algebraic equation solvers implemented in Casadi). Moreover, Pytorch [76] was employed for the actor and critic networks' definition and offline training.

**Table 2.** The MADDPG algorithm's hyperparameters.

| Hyperparameters | Value |
|---|---|
| MADDPG | |
| Discount factor ($\gamma$) | 0.99 |
| Batch size ($K$) | 50 |
| Buffer ($D$) | 5000 |
| Episodes ($N$) | 8000 |
| Time constant ($\kappa$) | 0.005 |
| Number of agents ($Na$) | 4 |
| $\eta, \beta, \varphi$ | (0.1, 0.1, 1) |
| Actor Network | |
| Activation function | ReLU, Tanh |
| Layers ($La_a$) | 4 |
| Neurons ($Nac_i$) | (200, 150, 150, 120) |
| Critic Network | |
| Activation function | ReLU, Linear |
| Layers ($La_c$) | 2 |
| Neurons ($Nc_i$) | (250, 150) |
| DNN Training Algorithm | |
| Optimizer | Adam |
| Actor learning rate ($\alpha_a$) | 0.0035 |
| Critic learning rate ($\alpha_c$) | 0.035 |
| Decay learning rate ($\vartheta$) | 0.1 |

The offline learning of the four agents following the RL framework is illustrated in Figures 8 and 9. In Figure 8, all agents resulted in an actor that maximized the policy gradient, having a magnitude compatible with that computed by the critic network (Figure 9) and independent of the initial conditions of the reactor. This means that all agents learned decision-making policies that maximized the rewards over the episodes. At this point, it was expected to obtain an actor and a critic sufficiently generalizing and robust to adapt to new process conditions. However, Agent 3 had an oscillatory behavior, which can make its online use unfeasible, and Agent 4 may have reached a sub-optimal state.

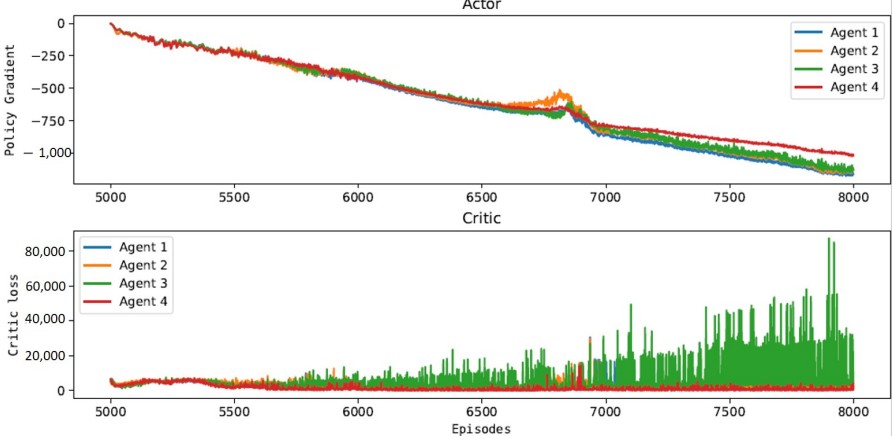

**Figure 8.** Actor and critic update to Van de Vusse reactor economic optimization and control.

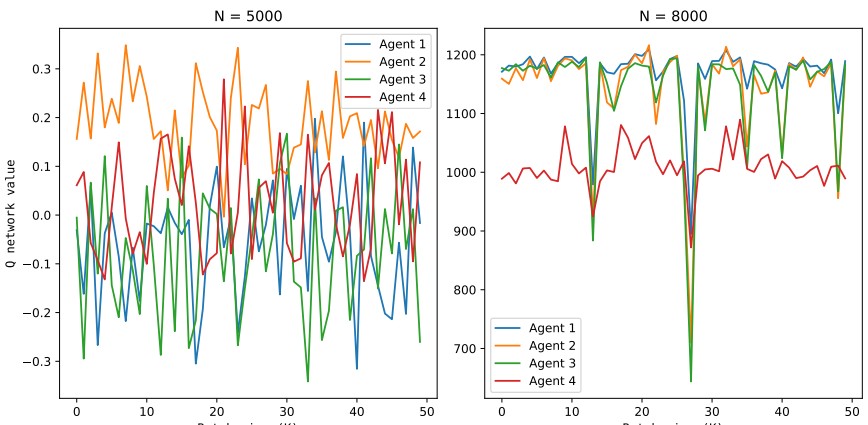

**Figure 9.** Q-network value considering the action taken from a continuous uniform distribution ($N = 5000$) and the action with the MADDPG algorithm ($N = 8000$).

Validation for Process Condition 1

To validate these results, a new process condition not tested offline was performed: the training time was extended to 5 h; the dynamics of process disturbance was a sine wave, which exhibited a smooth, periodic oscillation (Figure 10); the steady-state condition was $C_a(0) = 10$, $C_b(0) = 1.15$ and $F(0) = 150$.

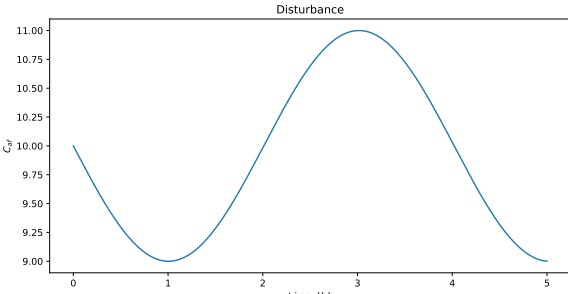

**Figure 10.** Process disturbance with sine wave dynamics outside the range used in the offline training.

In Figure 11, when evaluating the dynamics of $C_b$, Agents 1 and 4 had the least-oscillatory behavior, did not exceed the process constraint, and took adequate control actions for process disturbances, as illustrated in Figure 12. These results corroborate what was expected given the oscillatory behavior of Agents 2 and 3 in the offline training phase to update the actor and critic networks. Agent 3 resulted in a sub-optimal actor with saturated control actions at the upper limit of $F$. Agent 2 did not find a suitable actor to deal with process disturbances, resulting in an oscillatory and divergent control policy. These results denote one of the main features of the MADDPG algorithm (mentioned in [77]), as it allows using cooperative control agents that have increased learning (Agents 1 and 4) when collecting experiences of sub-optimal policies (Agents 2 and 3). Although Agents 2 and 3 resulted in unfeasible policies, the possibility of using each agent in parallel improved the robustness of the methodology (compared to DDPG), especially considering its implementation in the real process, which could discard Agents 2 and 3 and directly employ Agents 1 and 4 or even combine them. These claims were evaluated by comparing the performance of all agents with respect to the two-layer approach. Specifically, the HRTO approach with the set of parameters based on Matias and Le Roux [14] was chosen. Agents 1 and 4 rejected the disturbances and reached the set-point as in HRTO (Figure 13), which employed the extended Kalman filter (EKF) and estimated $C_{af}$ (online) for the model adaptation step of RTO and MPC. The EKF suitably estimated the disturbance (Figure 13b) so that MPC achieved the set-points resulting from the RTO layer (Figure 13a,c).

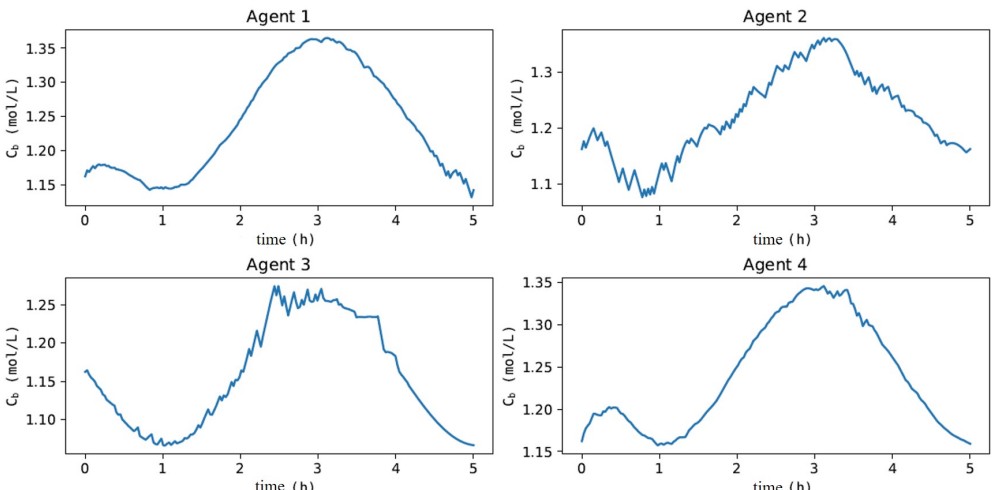

**Figure 11.** Dynamics of $C_b$ when subjected to process disturbances of Figure 10 over a 5 h control experiment.

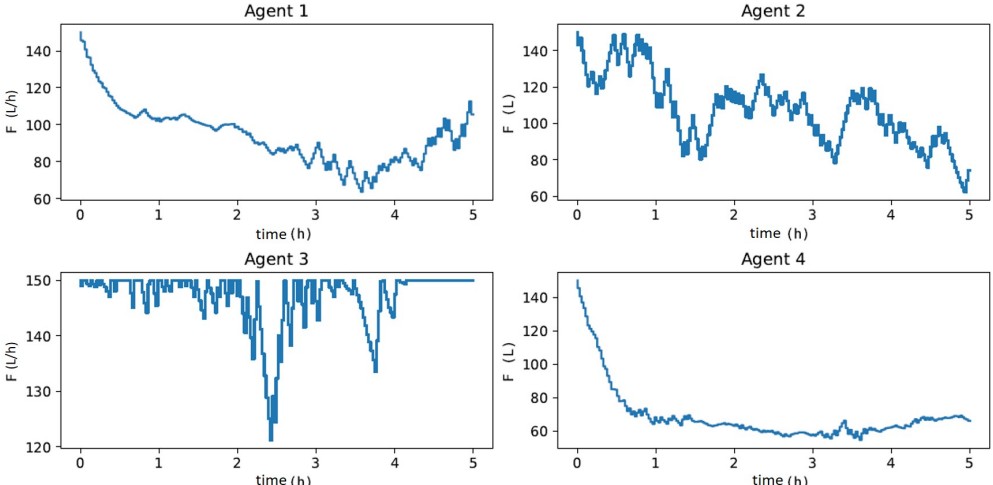

**Figure 12.** Dynamics of control actions ($\Delta F$) when subjected to process disturbances of Figure 10 over a 5 h control experiment, to maximize $C_b$.

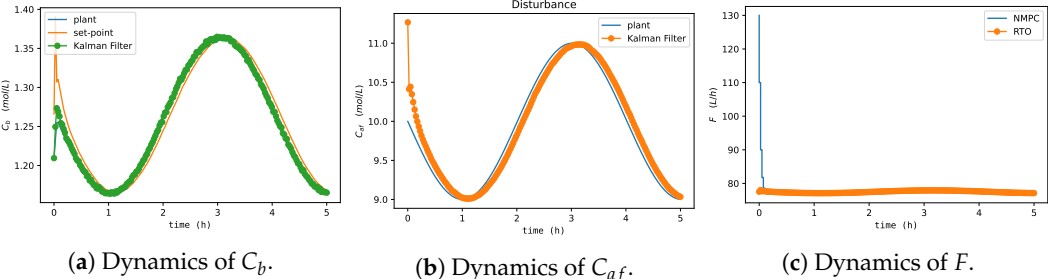

(**a**) Dynamics of $C_b$.      (**b**) Dynamics of $C_{af}$.      (**c**) Dynamics of $F$.

**Figure 13.** HRTO approach to Van de Vusse reactor economic optimization and control concerning process disturbances of Figure 10.

In Table 3, the average yield of $C_b$ and the CPU time for each agent are summarized. Agents 1 and 4 were an interrelated alternative to HRTO as they demand to compute a function rather than solve a sequence of nonlinear optimization problems, despite less economic profit. These results were also seen in Powell et al. [33]. However, using multiple agents guarantees learning from sub-optimal policies and parallel implementation.

**Table 3.** Average yield and online experiment time for each agent.

| Agent | Average Yield | Online Experiment Time |
|:---:|:---:|:---:|
| 1 | 1.2414 | 1.5 s |
| 2 | 1.2015 | 1.5 s |
| 3 | 1.1508 | 1.5 s |
| 4 | 1.2405 | 1.5 s |
| HRTO | 1.2534 | 16 s |

Validation for Process Condition 2

Another process condition is proposed to evaluate the generalization of agents when subjected to process disturbances that they were not exposed to during offline training, i.e., $C_{af}(t)$ has process dynamics according to Figure 14.

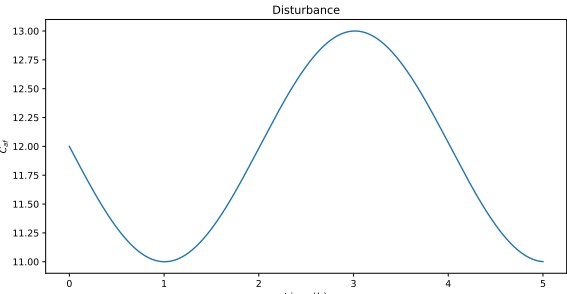

**Figure 14.** Process disturbance with sine wave dynamics outside the range used in the offline training.

All learning agents showed a more oscillatory behavior, as shown in Figure 15. This was expected as they were not exposed to these conditions in offline training. Despite this, they managed to capture the dynamic behavior of $C_b$ adequately, again with Agents 1 and 4 taking control actions closer to those considered optimal, as shown in Figure 16 (i.e., concerning HRTO (Figure 17)) and with a CPU time equal to the condition tested before and summarized in Table 3. These results were due to the networks' ability to generalize to other process conditions, which demonstrated the robustness of the MADDPG controller to adapt to abrupt process changes, as detailed in Spielberg et al. [58] for the DDPG controller, without the need to restart the offline training.

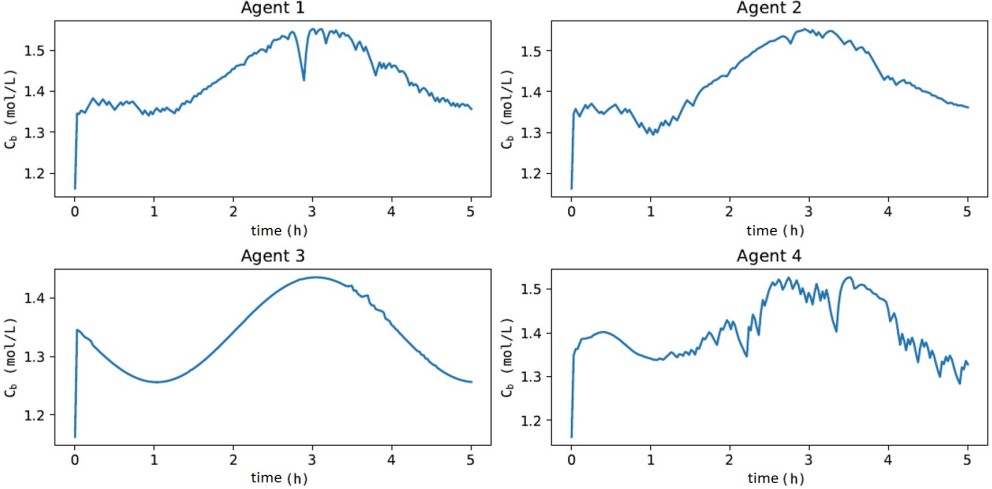

**Figure 15.** Dynamics of $C_b$ when subjected to process disturbances of Figure 14 over a 5 h control experiment.

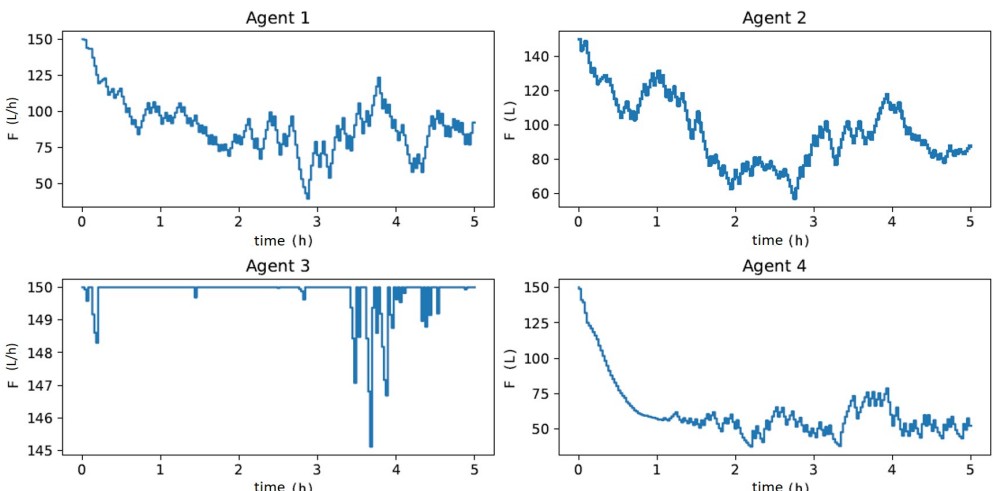

**Figure 16.** Dynamics of control actions ($\Delta F$) when subjected to process disturbances of Figure 14 over a 5 h control experiment, to maximize $C_b$.

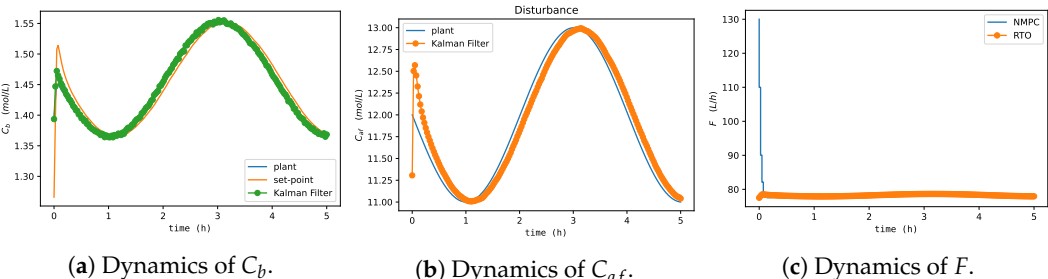

(**a**) Dynamics of $C_b$.      (**b**) Dynamics of $C_{af}$.      (**c**) Dynamics of $F$.

**Figure 17.** HRTO approach to Van de Vusse reactor economic optimization and control concerning process disturbances of Figure 14.

## 5. Conclusions

This review outlined guidelines for real-time optimization using reinforcement learning approaches. The analysis of specific applications for real-time optimization, supervisory control, and regulatory control allowed us to make some general conclusions about it:

- There are a huge number of RL applications not considering the economic optimization of the plant;
- Almost all applications are restricted to validation with bench-scale control experiments or based on simulation;
- There is a consensus in the literature that extensive offline training is indispensable to obtain adequate control agents regardless of the process;
- The definition of the reinforcement signal (reward) must be rigorously performed to adequately guide the agents' learning, which must be penalized when it is far from the condition considered ideal or when it results in impossible or unfeasible state transitions;
- The benchmark study of RL confirmed the hypothesis that cooperative control agents based on the MADDPG algorithm (i.e., one-layer approach) could be an option for the HRTO approach;
- Learning with cooperative control agents improved the learning rate (Agents 1 and 4) through the collection of experiences of sub-optimal policies (Agents 2 and 3);
- The parallel implementation with MADDPG is possible;
- The benefits of the collection of experiences with MADDPG depend on a trustworthy process simulation;
- Learning with MADDPG is fundamentally more difficult than the single agent (DDPG), especially for large-scale processes due to the dimensionality problem;

- It is necessary to develop RL algorithms to handle security constraints to ensure control stability and investigate applications for small-scale processes.

**Author Contributions:** As the article's main author, R.d.R.F. participated in all steps of the research method: conceptualization, methodology, writing—original draft preparation. Review and editing, all authors; conceptualization and supervision, B.D.O.C., A.R.S. and M.B.d.S.J. All authors have read and agreed to the published version of the manuscript.

**Funding:** This study was financed in part by the Coordenação de Aperfeiçoamento de Pessoal de Nível Superior—Brasil (CAPES)—Finance Code 001. Professor Maurício B. de Souza Jr. is grateful for the financial support from CNPq (Grant No. 311153/2021-6) and Fundação Carlos Chagas Filho de Amparo à Pesquisa do Estado do Rio de Janeiro (FAPERJ) (Grant No. E-26/201.148/2022).

**Data Availability Statement:** Not applicable.

**Conflicts of Interest:** The authors declare no conflict of interest.

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
