# Peer review of "One-Layer Real-Time Optimization Using Reinforcement Learning: A Review with Guidelines"

_processes, doi:10.3390/pr11010123_

Round 1

Reviewer 1 Report

The authors describe in their review the current situation and some applications for an interesting new method for control purposes. In general, an exciting paper and well written, however, in some places the authors could go into more detail.

My comments are as follows:

1.      Figure 1 needs an „adapted from [1]”

2.      Line 18 unclear statement: why have they evolved - because they have been integrated? In what direction did they evolve?

3.      Section lines 24 – 35: the entire paragraph must be reformulated for easier text understanding, it is written unnecessarily complex

4.      Line 33: why & how is it unfeasible? Pls give an example.

5.      Section line 36 – 48: are EMPC, DRTO, HRTO all equally good? Pros and Cons would be good to be given here. Are DRTO, HRTO also a one-layer approach?

6.      Figure 2 “right” instead of rigth

7.      Figure 4 needs a more detailed description, can the on-policy also be integrated?

8.      Eq. 11 is not well explained in the article and must be placed close too the first mentioning. Pls give an example or a better illustration.

9.      Figure 5 needs a more detailed description, particularly in the title.

10.  Section lines 268 – 271: what is the conclusion of Recht’s idea? Pls explain Eq. 16 as well in this circumstance.

11.  Lines 311 – 312: Either provide more information about the results of the last sentence or leave it out. What are the settings and the consequences of the control experiment?

12.  Lines 385 – 393: Why do these results occur, not just a description of what happens, also more details on figure 12 and the reasons Kalman filter was used, how it did improve the results etc.  would be good

13.  Lines 402 – 407: the authors present an extremely short description of the three figures. No reasons for results are stated, pls explain more for the interested reader.

14.  the Van de Vusse reactor is very prominently presented in the contribution. However, its setup is barely described to the readers and must be better explained.

15.  in line 425 -426, the last bullet point in the conclusions, it stays unclear what is meant with the development of RL algorithms, pls be more specific on process constraints and industrial applications

Reviewer 2 Report

The results presented in this paper seem correct. However, there are major comments from this reviewer to the authors. Even there is no major technical issue, however the most critical issue consists in the lack of any significant contribution for a possible publication in this current form.

1) Although the dissemination part is well described, but the novelty of the work in terms of theoretical developments is obscure, this issue should be elaborated in detail.

2) The Algorithm 1 may not be visual. The authors should give details of the explanation.

3) The disadvantage or the limitation of the proposed method must be described in conclusion.

4) All assumptions and constraints should be discussed.

5) How do you ensure the comparisons are fair and how the parameters set? Also, How do you ensure the results are enough to verify the proposal?

6) The authors should refer any further developments about the proposed approach and also refer any application of this strategy. Both topics should be discussion in the conclusion section.

7)       Check some typo and English representation. 

Round 2

Reviewer 2 Report

The article contains some novel part and the article has been revised according to reviewer's previous comment and suggestion. I have no further comment about this manuscript. I think that the paper deserves to be published.